# Identification of New Microfoci and Genetic Characterization of Tick-Borne Encephalitis Virus Isolates from Eastern Germany and Western Poland

**DOI:** 10.3390/v16040637

**Published:** 2024-04-19

**Authors:** Nina Król, Lidia Chitimia-Dobler, Gerhard Dobler, Dorota Kiewra, Aleksandra Czułowska, Anna Obiegala, Joanna Zajkowska, Thomas Juretzek, Martin Pfeffer

**Affiliations:** 1Institute of Animal Hygiene and Veterinary Public Health, University of Leipzig, 04103 Leipzig, Germanypfeffer@vetmed.uni-leipzig.de (M.P.); 2Department of Veterinary and Animal Sciences, University of Copenhagen, 1870 Frederiksberg, Denmark; 3Clinical Center for Emerging and Vector-Borne Infections, Odense University Hospital, 5000 Odense, Denmark; 4Bundeswehr Institute of Microbiology, 80937 Munich, Germany; 5Fraunhofer Institute of Immunology, Infection and Pandemic Research, 80799 Munich, Germany; 6Department of Parasitology, Institute of Biology, University of Hohenheim, 70599 Stuttgart, Germany; 7Department of Infectious Diseases and Tropical Medicine, Ludwig-Maximilians-University Munich, 80336 Munich, Germany; 8Department of Microbial Ecology and Acaroentomology, University of Wrocław, 51-148 Wrocław, Poland; 9Department of Infectious Diseases and Neuroinfections, Medical University in Białystok, 15-089 Białystok, Poland; joanna.zajkowska@umb.edu.pl; 10Center for Laboratory Medicine, Microbiology and Hospital Hygiene, Carl-Thiem-Klinikum Cottbus gGmbH, 03048 Cottbus, Germany

**Keywords:** tick-borne encephalitis, *Orthoflavivirus encephalitidis*, ticks, *Ixodes ricinus*, *Dermacentor reticulatus*, patients, nonendemic, microfocus

## Abstract

(1) Background: Tick-borne encephalitis (TBE) is the most important tick-borne viral disease in Eurasia, although effective vaccines are available. Caused by the tick-borne encephalitis virus (TBEV, syn. *Orthoflavivirus encephalitidis*), in Europe, it is transmitted by ticks like *Ixodes ricinus* and *Dermacentor reticulatus*. TBEV circulates in natural foci, making it endemic to specific regions, such as southern Germany and northeastern Poland. Our study aimed to identify new TBEV natural foci and genetically characterize strains in ticks in previously nonendemic areas in Eastern Germany and Western Poland. (2) Methods: Ticks were collected from vegetation in areas reported by TBE patients. After identification, ticks were tested for TBEV in pools of a maximum of 10 specimens using real-time RT-PCR. From the positive TBEV samples, E genes were sequenced. (3) Results: Among 8400 ticks from 19 sites, *I. ricinus* (*n* = 4784; 56.9%) was predominant, followed by *D. reticulatus* (*n* = 3506; 41.7%), *Haemaphysalis concinna* (*n* = 108; 1.3%), and *I. frontalis* (*n* = 2; <0.1%). TBEV was detected in 19 pools originating in six sites. The phylogenetic analyses revealed that TBEV strains from Germany and Poland clustered with other German strains, as well as those from Finland and Estonia. (4) Conclusions: Although there are still only a few cases are reported from these areas, people spending much time outdoors should consider TBE vaccination.

## 1. Introduction

Tick-borne encephalitis (TBE) is the most important tick-borne viral disease in Europe and Asia. It is caused by the tick-borne encephalitis virus (TBEV, syn. *Orthoflavivirus encephalitidis*) (Flaviviridae: Orthoflavivirus) [1]. TBEV can be divided into five subtypes, including the European, Siberian, Far Eastern, Baikalian, and Himalayan subtypes [2]. In Europe, two of these are circulating—the Siberian subtype, causing Russian spring–summer encephalitis, which is transmitted mostly by *Ixodes persulcatus,* and the European subtype, which is widely spread over the western continent and mainly associated with *I. ricinus* and *Dermacentor reticulatus* ticks [2,3].

Unlike many other tick-borne pathogens (TBPs), which are widespread in the tick population, TBEV circulates in very strictly defined, patchy natural foci. The identification of these microfoci—small geographical areas in which TBEV was detected in questing ticks at least once—is, therefore, difficult and many such areas remain undiscovered to date. Usually, microfoci are found based on data provided by TBE patients as an indication of potential locations in which tick bites were acquired [4]. Despite some limitations, the definition of risk areas, at least in Germany, helps formulate vaccination recommendations and monitor the spread of TBE. However, the TBE case incidence rate depends on temporal, climatic, and geographic factors, the vaccination coverage, the abundance of suitable tick vectors in the environment, and the population size of small mammals that are hosts of ticks and reservoirs for TBEV [5,6,7]. However, exposure risk is a very significant factor in harboring tick-borne diseases. One well-known risk factor for a TBEV infection is spending over 10 h per week in forests [8].

The virus is transmitted to humans through tick bites, although the possibility of infection by consuming unpasteurized dairy products has also been described [9,10,11,12]. In recent years, the TBE notification rate has increased in Europe, and the risk areas are expanding [13]. In Germany, TBE is endemic in the southern part, with a more recent contiguous northeastward spread into Saxony and a patchy spread to northwestern regions [14,15]. The number of reported cases in the last years varied from 400 to 700, and the federal states of Bavaria and Baden-Wuerttemberg account for more than 90% of them [4]. In Poland, the annual rate of diagnosed TBE cases is between 200 and 300, with the majority of cases being reported in the north-eastern part of the country (Warmian-Masurian and Podlaskie Voivodships), which is the endemic area; however, an increasing number of cases have been also noted in western parts of Poland [8,16].

In previous research based on phylogeographic analyses [17,18], it had been demonstrated that tick-borne flaviviruses, in general, follow an east-to-west spreading pattern and that the TBEV strains isolated in Germany have derived from ancestor strains mainly originating in Austria and the Czech Republic [19]. In our previous study, we isolated TBEV strains from northern Saxony, which showed very high similarity to isolates from Poland [20]. Two of these isolates originated from areas not known to be TBE endemic. This suggests an alternative migration way for the virus in comparison to the one across the Bavarian Forest originating from the Czech Republic and may suggest that the geographic transmission of TBEV from Poland is faster than from the endemic south of Germany.

The main objectives of this study were to (i) identify new TBE microfoci and (ii) isolate and genetically characterize TBEV strains from ticks collected in yet nonendemic areas from states and voivodeships that are adjacent to each other across the national border to better understand the geographic spread of TBEV in these areas.

## 2. Materials and Methods

### 2.1. Tick Collection

Ticks were collected in areas that TBE patients had indicated as putative places for the tick bites acquired. To achieve this, patients were approached in two ways. Firstly, data from patients was gathered by notifying local health authorities in Germany and Poland. Secondly, online surveys were conducted. In both cases, patients were asked to complete an anonymous questionnaire (TBE diagnosis date, tick bite date, and potential geographic location of tick bite). Some patients were able to identify the exact area of the tick bite, while others pointed out multiple locations. In total, based on the geographical information provided by patients, 19 potential TBE microfoci from Northeastern Germany (states: Brandenburg, Mecklenburg-West Pomerania, Saxony) and Western Poland (voivodeships: Lower Silesia, Lubush) could be selected for further investigations (Table 1, Figure 1a,b). In Saxony, four locations were selected, including Michalken, Sparte am Moor, Waldbad Bernsdorf, and Senftenberger See. In Mecklenburg-West Pomerania, one location was selected, Hiddensee Island. In Brandenburg, nine locations were chosen, including Cottbus, Frauendorf, Geierwalder See, Deulowitzer See, Göhlensee, Heidesee, Pinnower See, Spremberg See, and Löhsten. In Löhsten, 14 sub-spots (Figure 2) were indicated by a hunter who was diagnosed with TBE in Carl-Thiem-Klinikum, Cottbus. During his visit to the forest, he left his car only at these sub-spots to check on hunting. In Lubush, two sites were selected, incuding Jezioro Głębokie and Jeziory Wysokie. In Lower Silesia, there were three locations chosen—Dziewiętlin, Kamienna, and Marianówka. All three sites in Lower Silesia were indicated as locations of repeated tick bites near the places of residence. The authors were not able to obtain any patient data from Western Pomerania (Poland); thus, no flagging site was selected there.

At the chosen sites, ticks were collected from vegetation using a flagging method. Sites were visited from spring 2021 to summer 2022. Additionally, one partially engorged tick specimen was collected from a TBE patient at a hospital (Carl-Thiem-Klinikum Cottbus, Cottbus, Germany), and 41 non-engorged and partially engorged ticks were collected from a dog (Waldbad Bernsdorf, Germany). The collected ticks were identified life stage and species level [21,22] and stored at −80 °C until further tests.

### 2.2. Molecular Detection

Subsequently, due to the expected low prevalence of TBEV, ticks were tested in pools of a maximum of 10 specimens (per site, species, and life stage). Before RNA extraction, ticks were crushed with three rounds at a speed of 6.5 m/s for 30 s in the Fast Prep Savant FP120 tissue lyser (Bio101, Vista, CA, USA) in 1 mL minimum essential medium (MEM, Invitrogen, Karlsruhe, Germany) containing an antibiotic, antimycotic solution (ABAM, Invitrogen, Karlsruhe, Germany) [20]. Then, the nucleic acid was extracted using the MagNA Pure LC Total Nucleic Acid Kit (Roche, Mannheim, Germany) in the MagNA Pure LC instrument (Roche, Mannheim, Germany) according to the manufacturer’s instructions, with 200 μL of the tick homogenate supernatant. The extracted samples were tested for the presence of TBEV using a real-time RT-PCR (RT-qPCR) targeting the 3′-noncoding region of the viral genome and 5 μL of the eluted RNA [23]. For virus isolation, a 100 μL aliquot of the supernatants of each crushed RT-qPCR-positive tick pool was added to an 80% confluent cell culture of A549 cells (human lung carcinoma cells, German Collection of Microorganisms and Cell Cultures, DSMZ, Braunschweig, Germany). The homogenates were kept at −80 °C until they were used undiluted and in a dilution of 1:5 and 1:10 for virus isolation. After 1 h of incubation at 37 °C, the supernatant was decanted, and the cells were washed 5 times with MEM containing ABAM. A total of 5 mL of MEM containing 5-fold concentrated ABAM and 3% fetal calf serum were added. Cells were incubated for up to 7 days at 37 °C and observed daily for the occurrence of cytopathogenic effect (cpe). In the case of more than 50% cpe, the supernatant was taken and tested using RT-qPCR for TBEV, as described [23]. In case of no cpe, culture supernatant was taken after 7 days of incubation and also tested for the growth of TBEV using RT-qPCR. No subcultures were conducted. Based on previous experience, sequencing attempts directly from tick samples were performed only if the Ct value was <35. From the isolated TBEV strains, E genes were sequenced for confirmation, as described [20].

### 2.3. Statistical and Phylogenetic Analyses

To estimate the prevalence of TBEV infection in ticks, the minimum infection rate (MIR) was used, i.e., the minimum infected proportion expressed as a percentage [24], as follows:MIR = (p/N) × 100%(1)
where p  =  the number of positive pools, and N  =  the total number of ticks tested.

It was assumed that if a pool tested positive, only one tick specimen in this pool was infected. Fisher’s exact test was used to test for significant difference between two MIR estimates based on a two-tailed hypothesis [24]. To test for significant differences in pathogen pool prevalence between sites or life stages, we used a chi-square test without continuity correction [25]. The confidence intervals (95% CI), Fisher’s test, and chi-square test were performed using Graph Pad Prism Software v. 4.0. (Graph Pad Software Inc., San Diego, CA, USA). The significance threshold was set at *p* = 0.05.

The sequence data were processed using the program Geneious 9.1.5 (Biomatters, Auckland, New Zealand). A de novo assembly was conducted using the three chromatograms obtained from GATC (Eurofins Genomics, Ebersberg, Germany) for each positive sample. Nucleotides with an estimated error higher than 1% were trimmed. Subsequently, the sequences were cut to 1488 bp, the exact length of the envelope gene sequence. A ClustalW Alignment (GenomeNet, Kyoto University Bioinformatics Center, Kyoto, Japan) with several other E genes from selected isolates was performed, and a phylogenetic tree was generated using the PhyML 3.0 algorithm (Montpellier Bioinformatics, Montpellier, France) [26].

## 3. Results

A total of 8400 ticks were collected at 19 sites and from hosts (Table 2). Four tick species were identified, with *Ixodes ricinus* being the most widespread (*n* = 4784; 56.9%). The second most common species was *Dermacentor reticulatus* (*n* = 3506; 41.7%), followed by *Haemaphysalis concinna* (*n* = 108; 1.3%), and *I. frontalis* (*n* = 2; <0.1%). Nymphs were the predominant life stage in cases of *I. ricinus* (*n* = 3111; 65%) and *H. concinna* (*n* = 49; 45.4%). The second most abundant life stage for *H. concinna* were larvae (*n* = 32; 29.6%), while males (*n* = 14; 13%) and females (*n* = 13; 12%) were collected in almost equal numbers. For *I. ricinus*, females (*n* = 657; 13.7%; including one female collected from a patient at the hospital) and males (*n* = 634; 13.3%) were also found in almost equal proportions, and larvae (*n* = 382; 8%) were the least abundant life stage. In term of *D. reticulatus* ticks, females constituted 54.6% (*n* = 1915), and males constituted 45.4% (*n* = 1591). Within these, 41 were collected from a dog—13 females (31.7%) and 28 males (68.3%). Concerning *I. frontalis*, the two individuals were nymphs.

*Ixodes ricinus* ticks were found at almost all sites (Table 2, Figure 1), except for one in Brandenburg (Pinnower See, Germany), where only *D. reticulatus* ticks were collected. *Dermacentor reticulatus* was widely distributed in Brandenburg and Saxony (Germany), as well as in Lubush (Poland). It was absent, however, on an island in the northern part of Germany, Hiddensee Island (Mecklenburg, West Pomerania), and the southern part of Poland, a valley in the Sudetes mountain range, Marianówka, and Kamienna (Lower Silesia). *Haemaphysalis concinna* ticks were the most abundant in Saxony (all sites), and a few specimens were also found in Brandenburg and Lubush. Single individuals of *I. frontalis* were found at two sites in Saxony (Michalken and Senftenberger See), where all other tick species were collected, as well.

Overall, TBEV was detected in 19 tick pools, with a MIR of 0.23% (95% CI: 0.14–0.36). The virus was found in two tick species, *I. ricinus* (13 pools; MIR = 0.27%; 95% CI: 0.15–0.47) and *D. reticulatus* (six pools; MIR = 0.17%; 95% CI: 0.07–0.38), with no statistical differences (*p* = 0.486). In *I. ricinus*, TBEV was found in females (seven pools; MIR = 1.07%; 95% CI: 0.47–2.23), males (two pools; MIR = 0.32%; 95% CI: <0.01–1.22), and nymphs (four pools; MIR = 0.13%; 95% CI: 0.04–0.34), with MIR being significantly higher for females (χ^2^ = 16.18; df = 2; *p* < 0.01). In contrast, the virus was detected in females (two pools; MIR = 0.1%; 95% CI: <0.01–0.41) and males (four pools; MIR = 0.25%; 95% CI: 0.07–0.67) of *D. reticulatus* with no significant difference (*p* = 0.42).

Positive samples originated from six sites (Table 3, Figure 2, Figure 3, Figure 4, Figure 5, Figure 6 and Figure 7), five in Germany and one in Poland. The MIR for these TBEV-positive sites was 0.5% (95% CI:0.31–0.78). The highest MIR was observed in Löhsten (1.05%; 95% CI: 0.46–2.19) (Figure 2), followed by Göhlensee (0.95%; 95% CI: 0.38–2.11) (Figure 3), Senftenberger See (0.36%; 95% CI: <0.01–2.21) (Figure 4), Waldbad Bernsdorf (0.32%; 95% CI: <0.01–1.24) (Figure 5), Marianówka (0.23%; 95% CI: <0.01–0.88) (Figure 6), and Sparte am Moor (0.13%; 95% CI: <0.01–0.84) (Figure 7). However, the differences in MIR between the locations with positive pools were just above the significance level (χ^2^ = 10.454; df = 5; *p* = 0.063). None of the ticks collected from the dog or patient were TBEV-positive.

At Löhsten site, where MIR was the highest, a total of 669 ticks were collected from 14 sub-spots (Figure 2, Table 3). The dominant species there was *D. reticulatus* (96.6%; *n* = 646), followed by *I. ricinus* (2.4%; *n* = 16) and *H. concinna* (1%; *n* = 7). TBEV was detected in seven pools (MIR: 1.05%; 95% CI: 0.46–2.19) from two sub-spots (Figure 2, Table 4), including one pool of *I. ricinus* male, two pools of *D. reticulatus* females, and four pools of *D. reticulatus* males. At sub-spot no. 4, one pool was positive from 90 ticks collected (MIR: 1.1%; 95% CI: <0.01–6.6), and at sub-spot no. 10, six pools were positive from 160 ticks collected (MIR: 3.75%: 95% CI: 1.6–10.2). There were no significant differences in MIR between infected species *D. reticulatus* and *I. ricinus* (*p* = 0.158).

Unfortunately, virus cultivation was not successful for all positive pools, which may have been due to low virus load or the sample transport between different laboratories. From 19 positive pools, only twelve strains were successfully cultivated and sequenced, including ten from Germany—seven from Brandenburg, Löhsten (*n* = 5), Göhlensee (*n* = 2); three from Saxony, Sparte am Moor (*n* = 1) and Waldbad Bernsdorf (*n* = 2); and two from Poland, Lower Silesia and Marianówka. The phylogenetic analyses based on the E-gene revealed that all strains from Löhsten (Brandenburg) clustered the closest with German strains from Baden-Württemberg, Tübingen virus and Emmendingen (Figure 8). One sample from Göhlensee (Brandenburg) has the closest genetic relation to northeastern European strains from Finland and Estonia, and the second of the samples from the current study from Saxony. All isolates from Saxony (from both locations, Sparte am Moor and Walbad Bernsdorf) have the closest genetic relation to those from Baden-Württemberg, Wangen and Karsee. Samples from Poland (Lower Silesia, Marianówka) cluster the closest with German strains from North Rhine-Westphalia.

## 4. Discussion

The goals of the current study were the identification of new tick-borne encephalitis (TBE) microfoci, as well as the isolation and phylogenetic characterization of the tick-borne encephalitis virus (TBEV) strains circulating in nonendemic areas in Eastern Germany and Western Poland. Our strategy involved gathering information about potential locations from TBE patients who indicated the areas where they had most likely acquired tick bites. Subsequently, ticks were sampled in those locations for TBEV to increase the success of finding TBE microfoci. It must be mentioned that the research was conducted in 2021 and 2022, and due to the COVID-19 pandemic, some local health authorities were overwhelmed, and the notification of new TBE cases was delayed. However, despite these limitations, our approach seems to be an effective way of searching for new microfoci, as we were able to indicate five potential sites in Poland and fourteen in Germany.

We collected four tick species from the vegetation, including *Ixodes ricinus*, *Dermacentor reticulatus*, *Haemaphysalis concinna* (in Germany and Poland), and *I. frontalis* (only in Germany). These findings are in line with the distribution ranges of these species [21,27,28,29]. Interestingly, *I. frontalis*, an ectoparasite of birds that is usually collected from avian hosts, had been found on vegetation previously, also in Germany [30].

TBEV was detected in two tick species, *I. ricinus* and *D. reticulatus*, that are known vectors of TBEV in Europe [2,20,31,32]. The overall minimal infection rate (MIR) of TBEV was low, 0.23%, with no significant differences between the two species. A previous study from a nonendemic area in Germany [20] had also shown no statistical difference in MIRs between *I. ricinus* and *D. reticulatus*, despite slightly higher rates (0.57% and 0.59%, respectively). It is suggested that *D. reticulatus* contributes, besides *I. ricinus*, to the virus expansion and helps to maintain circulation in new microfoci [31,33].

First-time virus detection in questing ticks was successful at six out of nineteen investigated locations. MIRs varied between microfoci from 0.13% to 1.05% and between *I. ricinus* life stages from 0.13% (nymphs) to 1.07% (females), which represents the general view of TBEV prevalence from previous European studies [24,34,35,36,37,38,39,40]. Additionally, we tested ticks removed from two hosts. The tick removed from the TBE patient was a partially engorged *I. ricinus* female, and from the dog (Waldbad Bernsdorf, Bernsdorf, Germany), 41 *D. reticulatus* ticks were removed (13 females and 28 males, all non- or partially engorged). However, despite a confirmed diagnosis, no TBEV could be amplified from the patient’s tick or from the dog walking in the area where two pools were TBEV positive. TBE viremia is known to be short-lived and of low titer, which is why humans are considered dead-end hosts and attempts to detect TBEV in attached ticks usually fail [41].

We were able to identify six previously unknown microfoci in areas where human TBE cases have been reported, five in Germany (two in Brandenburg and three in Saxony) and one in Poland (Lower Silesia). Detailed coordinates of the microfoci are available in Table 1. Of these microfoci, three are located in districts that have recently been registered as TBE risk areas—Göhlensee (Oder-Spree County since 2022), Waldbad Bernsdorf, and Sparte am Moor (Bautzen County since 2018). The distance between these three microfoci is within 17 km, making it probable that the virus has spread directly from one microfocus to another through tick-infested or viremic hosts, such as mammals or birds [42]. The microfocus Marianówka in Poland is located in Kłodzko County, which has the highest number of human cases (4–10 per year) in the Lower Silesia Voivodship (10–18 per year). The Löhsten microfocus with the highest MIR is situated less than 30 km away from a previously described TBEV microfocus (Battaune) [20]. In Löhsten, TBEV was detected in two sub-sites (no. 4 and 10) that are 1.4 km apart. However, Löhsten and Battaune are in two different districts, and neither are considered TBE risk areas according to the federal Robert Koch Institute. Consequently, vaccination against TBE is not recommended in these areas, even though cases occur, and we have detected the virus presence there.

The TBEV sequences obtained from Löhsten (Brandenburg) show a close phylogenetic relation with strains from endemic areas in Baden-Württemberg, Tübingen virus and Emmendingen (located more than 500 km to the southwest), and the strains circulating in subsite no. 10 (not larger than 0.5 ha) most likely originated from two sources. There is no evidence that the isolates from Löhsten are related to those from the natural focus in Battaune, which are clustering with sequences from Poland and Bavaria (Neustadt an der Waldnaab, Germany) [20]. TBEV in Göhlensee (Brandenburg)—the southeastern side of the lake—has also probably been introduced into the area at least twice independently. One sample has the closest genetic relationship with northeastern European strains from Finland and Estonia, which were most likely carried by birds, and the second one with the samples from microfoci that are located 70 km (Waldbald Bernsdorf) and 80 km (Sparte am Moor) to the south and could have been transported by ticks feeding on mammals, e.g., deer or wild boar, as well as birds [43,44]. This suggests that two different strains are circulating in this location. Isolates from Saxony, microfoci Sparte am Moor and Walbad Bernsdorf, which are about 10 km apart, are closely related to each other, and the strains show the closest clustering with those from Baden-Württemberg, Wangen and Karsee (over 500 km south-west), implying a spread by migratory birds. Unfortunately, sequencing of the sample from the third microfocus in Saxony, Senftenberger See, was not successful. The TBEV sequences from Marianówka (Poland) show that the virus circulating in this microfocus is phylogenetically closely related to German strains from North Rhine-Westphalia. These strains are more than 600 km away in a westerly direction.

As for the rest of the sites where no TBEV-positive ticks were found, it is likely that the patients did not correctly recall where they had acquired tick bites or that we missed the right location when flagging. It is also very common for patients suffering from TBDs not to remember being bitten by a tick [45,46].

## 5. Conclusions

In summary, our results show that the patient-derived approach is successful in the search for new TBEV microfoci. The TBEV prevalence (MIR) within the study area was rather low. The detection of previously unknown TBEV microfoci highlights the risk of TBE even in nonendemic areas. Increased public awareness regarding vaccination and surveillance efforts are therefore required. The phylogenetic relationships presented for the newly isolated TBEV strains from Eastern Germany and Southwestern Poland indicate that it is not a simple geographical spread that links areas with TBEV outbreaks. In particular, the significance of the noncontinuous distribution of TBEV patterns over long distances is surprising. Genetic data suggested, among other things, an association with bird migration.

## Figures and Tables

**Figure 1 viruses-16-00637-f001:**
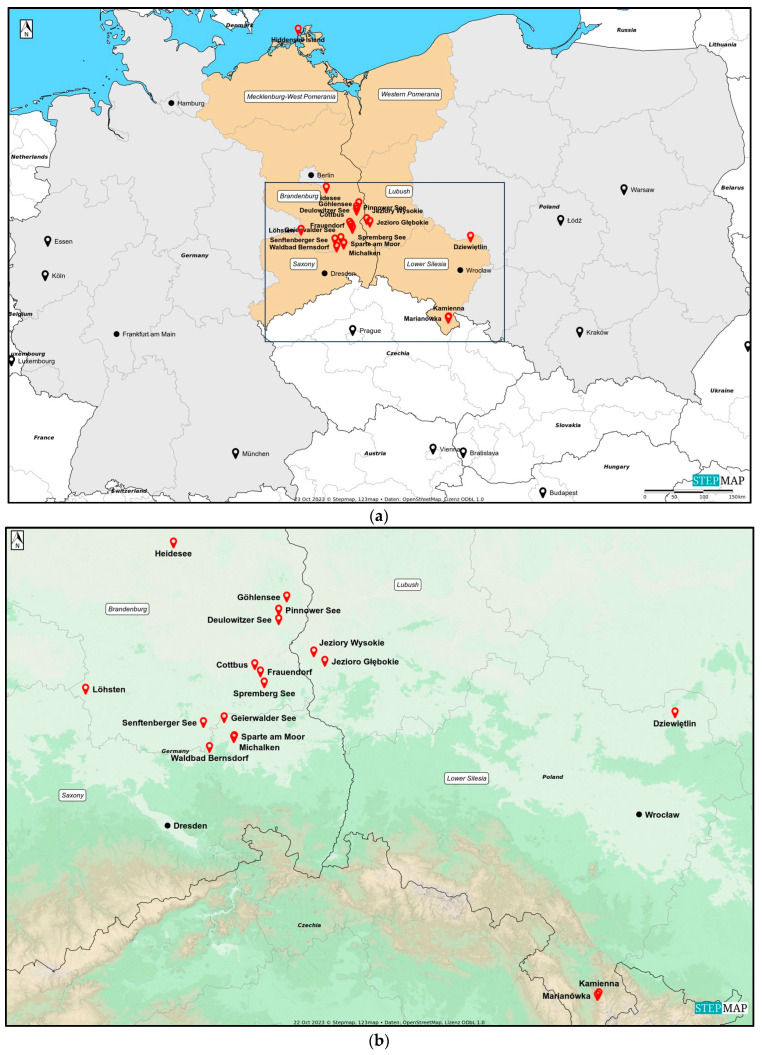
Tick collecting sites in Germany and Poland indicated by tick-borne encephalitis (TBE) patients (© StepMap, 123 map, Data: OpenStreetMap, License ODbL 1.0); (**a**) German states and Polish voivodships where the study was conducted (orange); (**b**) collection sites selected for the study (red pins).

**Figure 2 viruses-16-00637-f002:**
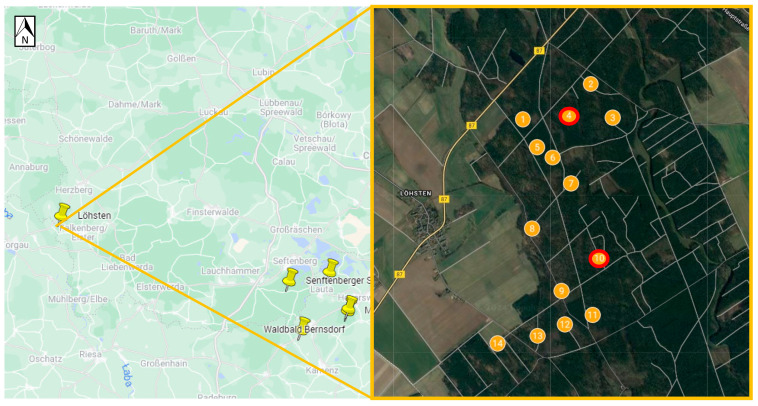
Site Löhsten (Germany, Brandenburg). Ticks were collected at 14 sub-spots indicated by a hunter. Tick-borne encephalitis virus was found at two sub-spots (red circle)—one pool at sub-spot no. 4 and six pools at sub-spot no. 10 (Google Maps with own modifications).

**Figure 3 viruses-16-00637-f003:**
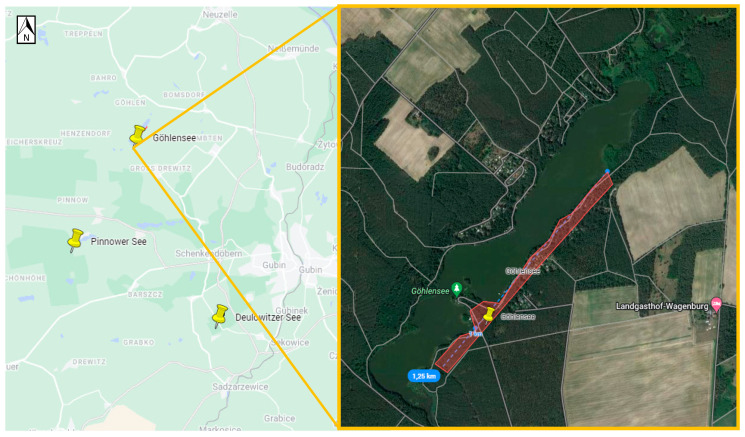
Site Göhlensee (Germany, Brandenburg). The flagging area is marked in transparent red, where six positive pools were found (Google Maps with own modifications).

**Figure 4 viruses-16-00637-f004:**
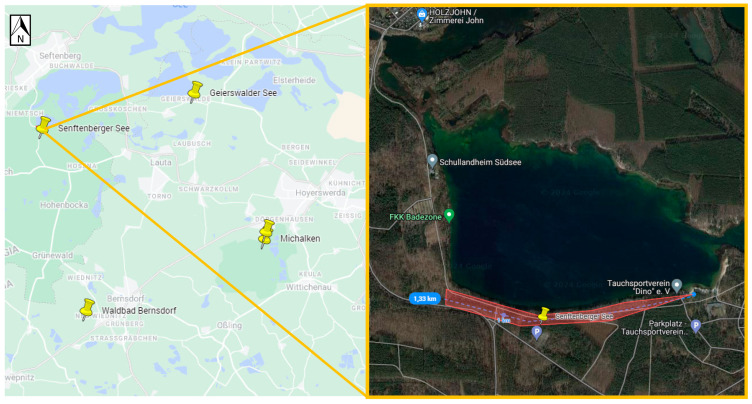
Site Senftenberger See (Germany, Saxony). The flagging area is marked in transparent red, where one positive pool was found (Google Maps with own modifications).

**Figure 5 viruses-16-00637-f005:**
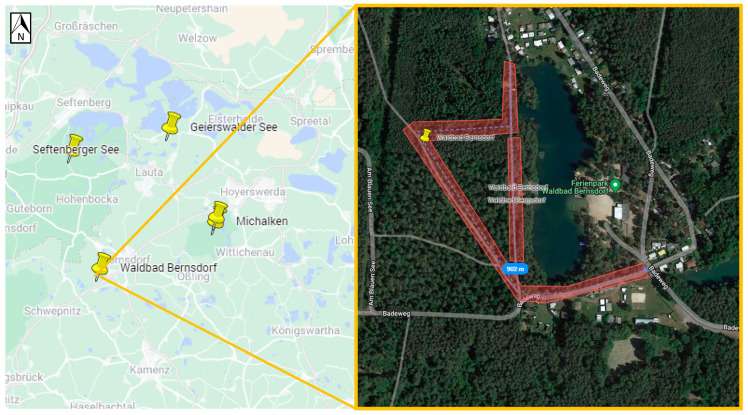
Site Waldbad Bernsdorf (Germany, Saxony). The flagging area is marked in transparent red. where two positive pools were found (Google Maps with own modifications).

**Figure 6 viruses-16-00637-f006:**
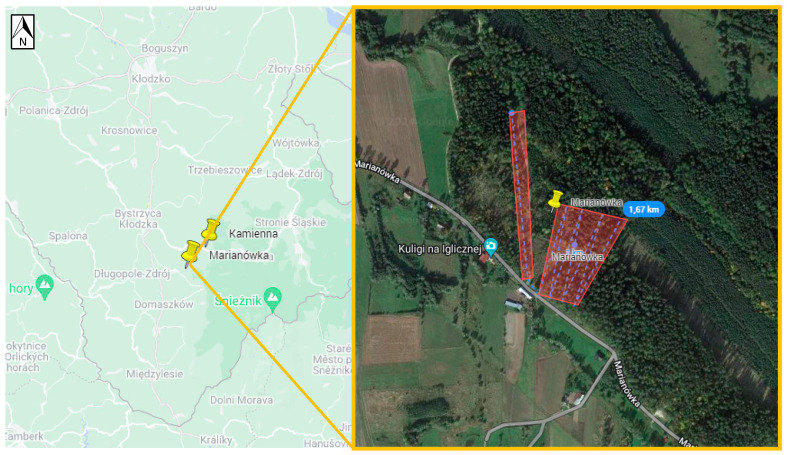
Site Marianówka (Poland, Lower Silesia). The flagging area is marked in transparent red, where two positive pools were found (Google Maps with own modifications).

**Figure 7 viruses-16-00637-f007:**
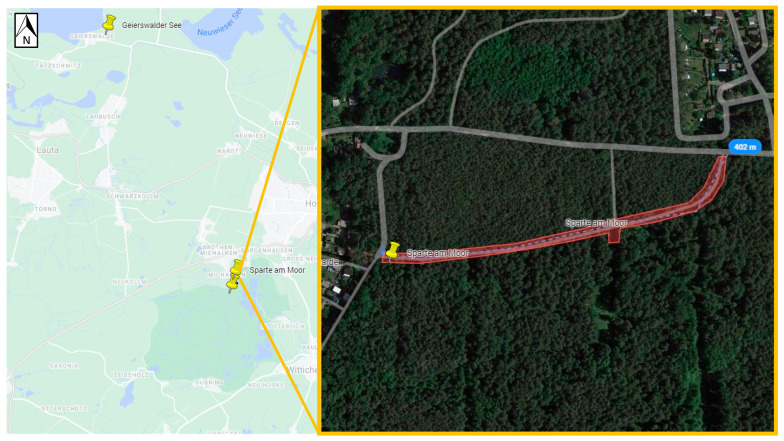
Site Sparte am Moor (Germany, Saxony). The flagging area is marked in transparent red, where one positive pool was found (Google Maps with own modifications).

**Figure 8 viruses-16-00637-f008:**
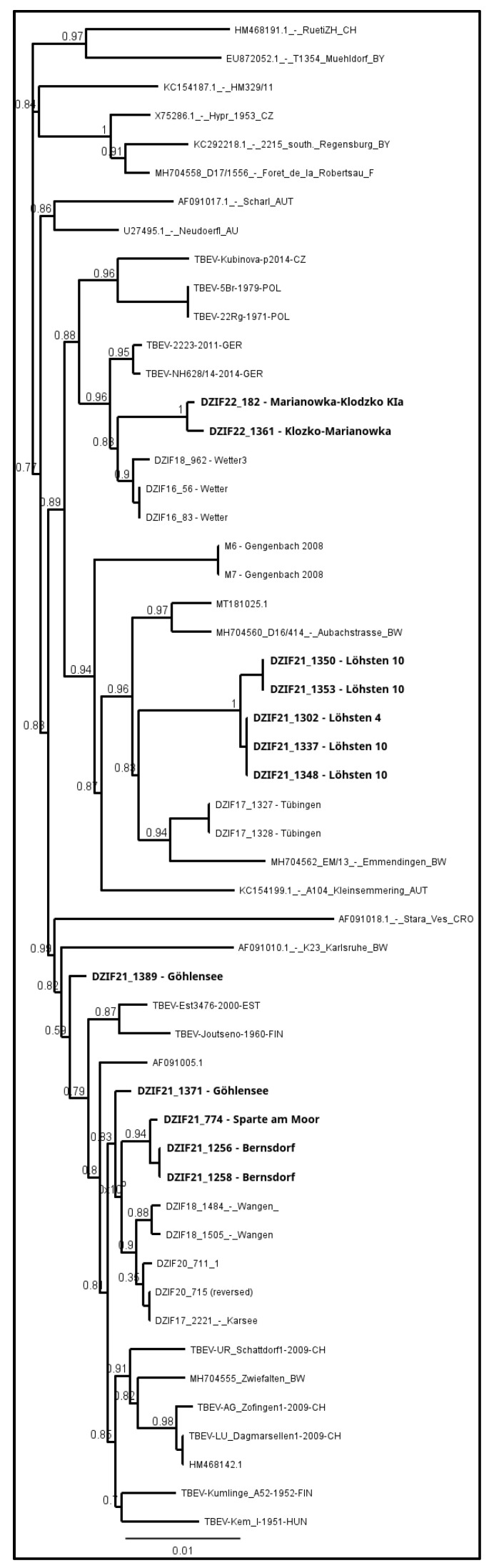
Phylogenetic tree for tick-borne encephalitis virus (TBEV) based on E-gene sequences. Samples from this study are marked in bold.

**Table 1 viruses-16-00637-t001:** Location and description of 19 potential tick-borne encephalitis (TBE) microfoci selected for investigations.

Country	State/Voivodship	Name of Site	GPS Coordinates	Short Description
Germany	Brandenburg	Cottbus	51.72912, 14.34782	the forest between a settlement and the lake
		Frauendorf	51.69781, 14.38748	woodlands near the residential area
		Geierwalder See	51.49863, 14.13382	green areas between lakes
		Deulowitzer See	51.92312, 14.64813	the forest around the lake
		Göhlensee	52.02116, 14.57116	the walking path on the eastern side of the lake
		Heidesee	52.25275, 13.77984	the forested area between lakes
		Löhsten	51.62287, 13.16589	14 sub-spots in the forest
		Pinnower See	51.96456, 14.51513	woodlands between the lake and the settlement
		Spremberg See	51.64993, 14.41396	green areas around the lake
	Mecklenburg-West Pomerania	Hiddensee Island	54.53988, 13.09666	the walking area on the island
	Saxony	Michalken	51.41198, 14.20098	the forest near residential areas
		Senftenberger See	51.47759, 13.98911	the walking path along the southern part of the lake
		Sparte am Moor	51.41669, 14.20562	the walking path in a forest
		Waldbad Bernsdorf	51.36904, 14.03123	walking paths in the forest around the small lake
Poland	Lubush	Jezioro Głębokie	51.74357, 14.8375	the forest around the lake
		Jeziory Wysokie	51.78434, 14.76023	walking paths in the forest
	Lower Silesia	Kamienna	50.28478, 16.75311	green areas around the settlement
		Marianówka	50.26388, 16.72314	the forest close to the settlement
		Dziewiętlin	51.44850, 17.31076	walking paths in the forest

**Table 2 viruses-16-00637-t002:** The number of collected ticks at 19 flagging sites and from 2 hosts (the tick-borne encephalitis patient at a hospital and the dog from one of the flagging sites).

Country	State/Voivodship	Name of Site	Number of Collected Ticks				Total
			*D. reticulatus*	*H. concinna*	*I. frontalis*	*I. ricinus*	
Germany	Brandenburg	Cottbus	0	0	0	76	76
		* Carl-Thiem-Klinikum Cottbus (from a patient)	0	0	0	1 *	1 *
		Frauendorf	3	0	0	223	226
		Geierwalder See	18	0	0	34	52
		Deulowitzer See	2	0	0	21	23
		Göhlensee	10	2	0	620	632
		Heidesee	0	0	0	62	62
		Löhsten	646	7	0	16	669
		Pinnower See	124	0	0	0	124
		Spremberg See	6	0	0	33	39
	Mecklenburg-West Pomerania	Hiddensee Island	0	0	0	205	205
	Saxony	Michalken	163	60	1	207	431
		Senftenberger See	77	6	1	195	279
		Sparte am Moor	114	26	0	604	744
		Waldbad Bernsdorf	278	4	0	344	626
		* Waldbad Bernsdorf (from a dog)	41 *	0	0	0	41 *
	Subtotal		1482	105	2	2641	4230
Poland	Lubush	Jezioro Głębokie	1928	2	0	67	1997
		Jeziory Wysokie	58	1	0	76	135
	Lower Silesia	Kamienna	0	0	0	123	123
		Marianówka	0	0	0	880	880
		Dziewiętlin	38	0	0	997	1035
	Subtotal		2024	3	0	2143	4170
Total			3506	108	2	4784	8400

* Ticks collected from hosts.

**Table 3 viruses-16-00637-t003:** Tick-borne encephalitis virus-positive pools and minimum infection rates (MIRs) detected in ticks.

Country	State/Voivodship	Name of Site	Number of Positive Pools (MIR)					
			*D. reticulatus*		*I. ricinus*			Total
			Females	Males	Females	Males	Nymphs	
Germany	Brandenburg	Göhlensee	0	0	5 (2.99%)	1 (0.79%)	0	6 (0.95%)
		Löhsten	2 (0.66%)	4 (1.17%)	0	1 (33.33%)	0	7 (1.05%)
	Saxony	Senftenberger See	0	0	0	0	1 (0.62%)	1 (0.36%)
		Sparte am Moor	0	0	0	0	1 (0.18%)	1 (0.13%)
		Waldbad Bernsdorf	0	0	1 (1.2%)	0	1 (0.55%)	2 (0.32%)
	Subtotal		2 (0.36%)	4 (0.71%)	6 (2.03%)	2 (0.75%)	3 (0.24%)	17 (0.58%)
Poland	Lower Silesia	Marianówka	0	0	1 (0.9%)	0	1 (0.19%)	2 (0.23%)
Total			2 (0.36%)	4 (0.71%)	7 (1.76%)	2 (0.54%)	4 (0.23%)	19 (0.5%)

**Table 4 viruses-16-00637-t004:** Detailed data on collected ticks (F—females, M—males, N—nymphs, L—larvae) and detected tick-borne encephalitis virus-positive pools from site Löhsten (Germany, Brandenburg).

Site	Number of Collected Ticks (Number of TBEV-Positive Pools)										
	*D. reticulatus*			*H. concinna*			*I. ricinus*				Total
	F	M	Subtotal	N	L	Subtotal	F	M	N	Subtotal	
Löhsten 1	39	39	76	1	0	1	0	0	0	0	77
Löhsten 2	22	22	44	0	0	0	0	0	1	1	45
Löhsten 3	6	8	14	0	0	0	0	0	0	0	14
Löhsten 4	36 (1)	53	89 (1)	0	0	0	0	1	0	1	90 (1)
Löhsten 5	6	6	12	0	0	0	0	0	0	0	12
Löhsten 6	17	25	42	0	0	0	1	0	0	1	43
Löhsten 7	18	28	46	0	1	1	0	0	0	0	47
Löhsten 8	24	28	52	0	0	0	0	0	1	1	53
Löhsten 9	12	5	17	0	0	0	0	1	0	1	18
Löhsten 10	81 (1)	78 (4)	159 (5)	0	0	0	0	1 (1)	0	1 (1)	160 (6)
Löhsten 11	0	3	3	3	0	3	0	0	4	4	10
Löhsten 12	1	0	1	0	0	0	0	0	1	1	2
Löhsten 13	33	40	73	2	0	2	0	0	5	5	80
Löhsten 14	10	8	18	0	0	0	0	0	0	0	18
Total	303 (2)	343 (4)	646 (6)	6	1	7	1	3 (1)	12	16 (1)	669 (7)

## Data Availability

Data may be available upon request.

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
