# Peer review of "Identification of New Microfoci and Genetic Characterization of Tick-Borne Encephalitis Virus Isolates from Eastern Germany and Western Poland"

_viruses, 2024, doi:10.3390/v16040637_

Round 1
Reviewer 1 Report
Comments and Suggestions for Authors
Line 121: 6.5 rpm seems to be quite slow. Please recheck the condition.
Line 137: How the authors determine the CPE is more than 50% in the infected cells?
-In Materials and Methods, the authors suggested that RT-qPCR were performed to detect TBEV from tick pools and positive homogenates inoculated cell culture supernatant. However, there are no information on quantity of TBEV. Authors mentioned that they could not isolate virus from some of the positive samples, due to the low load of virus in the samples. The authors should provide the viral load in those samples to support their hypothesis.
Line 269-270: It’s not only D. reticulatus but also I. ricinus that contributes to the TBEV expansion according to the data in this study.
Author Response
Dear Reviewer,
thank you for your remarks. We have addressed all your comments. You can see our responses below. We have also improved the English of our manuscript. All changes are marked in red in the text.
Line 121: 6.5 rpm seems to be quite slow. Please recheck the condition.
- Thank you for pointing this out. The sentence has been edited:
- “Before RNA extraction, ticks were crushed with three rounds at a speed of 6.5 for 30 s”
Line 137: How the authors determine the CPE is more than 50% in the infected cells?
- The typical cpe of TBE infection is a rounding up of usually adherent epithelial cells. If microscopically estimated >50% of epithelial A549 cells were rounded up or floating in the supernatant, it was taken for PCR testing. However, we did not add an explaining sentence in the manuscript text.
In Materials and Methods, the authors suggested that RT-qPCR were performed to detect TBEV from tick pools and positive homogenates inoculated cell culture supernatant. However, there are no information on quantity of TBEV. Authors mentioned that they could not isolate virus from some of the positive samples, due to the low load of virus in the samples. The authors should provide the viral load in those samples to support their hypothesis.
- Because qPCR is very sensitive, it can detect very low levels of genetic material in samples. However, in positive samples with a Ct value above 35, the amount of viral RNA is too low for sequencing. Therefore, we only sequence and try to isolate the virus only from the samples with a Ct below 35. The following sentence has been added to the “Molecular detection” section:
- “Based on previous experience, sequencing attempts directly from tick samples were performed only if the Ct value was <35.”
Line 269-270: It’s not only D. reticulatus but also I. ricinus that contributes to the TBEV expansion according to the data in this study.
- This sentence is based on cited references and does not refer only to our study, but to the general situation where D. reticulatus can maintain viral circulation, in the absence of I. ricinus. For clarification, the sentence has been changed in the manuscript as suggested:
- “It is suggested that reticulatus contributes, besides I. ricinus, to the virus expansion and helps to maintain circulation in new microfoci [31,33].”
Reviewer 2 Report
Comments and Suggestions for Authors
Dear Authors,
I enjoyed reading this publication. The work is interesting, based on rich material, and well-prepared editorially. In the future, it may be worth supplementing the results with a study of rodents, as well as ticks collected from hosts in these areas.
Minor editorial comments:
Page 1, line: 40, 41 - usually the family is mentioned first, just (Flaviviridae: Flavivirus)
Page 3, table 1: please harmonize as there is both "a forest around the lake" and "a forest around a lake"
Page 15, line 397; page16, 447: confusion with the numbering of references
Congratulations and best regards,
Author Response
Dear Reviewer,
thank you for your kind comments. We have addressed all your remarks. We have also improved the English of our manuscript. All changes are marked in red in the text. You can see our responses to your comments below.
Page 1, line: 40, 41 - usually the family is mentioned first, just (Flaviviridae: Flavivirus)
- As the nomenclature of the genus Flavivirus has changed to Orthoflavivirus (https://link.springer.com/article/10.1007/s00705-023-05835-1) as well the name of TBEV to Orthoflavivirus encephalitidis, we implemented respectful changes in the manuscript and references (changed the first position to Worku, D.A. Tick-borne encephalitis (TBE): from tick to pathology. J Clin Med 2023, 12, 6859, doi: 10.3390/jcm12216859). However, we decided to use TBEV as a synonym throughout the manuscript. The sentence in the manuscript has been changed according to the suggestion and update:
- “It is caused by tick-borne encephalitis virus (TBEV) (Flaviviridae: Orthoflavivirus) [1].”
- “It is caused by tick-borne encephalitis virus (TBEV) (Flaviviridae: Orthoflavivirus) [1].”
Page 3, table 1: please harmonize as there is both "a forest around the lake" and "a forest around a lake"
- Thank you for spotting this. All indefinite articles in the table have been changed to definite articles.
Page 15, line 397; page16, 447: confusion with the numbering of references
- Thank you for bringing this to our attention. The numbering of the references has been corrected (also ref. no. 8, 41, 45, and 46.
Reviewer 3 Report
Comments and Suggestions for Authors
The manuscript by Król and colleagues describes a large study focused on identifying new foci of tick-borne encephalitis virus in Germany and Poland. The manuscript is logically presented and generally well written, although copy editing would clearly enhance readability and eliminate several incomplete sentences and other grammatical errors.
A strength of the work described is using patient traceback to zero in on potential new areas of TBEV endemicity, several of which were identified. The sequencing data also is of value in placing these new isolates into phylogenetic perspective. Overall, this manuscript appears to provide meaningful and useful information that extends our knowledge of the range of this important pathogen. I do not have major criticisms, but offer a few comments for consideration:
This manuscript is exceptionally specific to geography and have doubts that the maps represented in Figures 2-7 are worth including. Similarly, Figure 1b is clearly of value, but 1a could be deleted. This should be the author’s choice but something to consider, especially if they have not obtained copyright permissions for those images.
The subheadings (e.g. 1) Background) do not seem standard for Viruses and should likely be removed.
Line 225-6: sentence requires editing.
There are some strange mutations in the reference list e.g., line 397 (2. 22. Siuda).
Comments on the Quality of English Language
A reasonable number of incomplete sentences and other sentence malformations.
Author Response
Dear Reviewer,
thank you for your valuable insights. We have addressed all your comments. We improved the English of our manuscript. All changes are marked in red in the text. You can see our responses to your comments below.
The manuscript by Król and colleagues describes a large study focused on identifying new foci of tick-borne encephalitis virus in Germany and Poland. The manuscript is logically presented and generally well written, although copy editing would clearly enhance readability and eliminate several incomplete sentences and other grammatical errors.
- Thank you for pointing this out. We have double-checked the manuscript for editing and grammar. Hopefully, all the errors have been corrected.
A strength of the work described is using patient traceback to zero in on potential new areas of TBEV endemicity, several of which were identified. The sequencing data also is of value in placing these new isolates into phylogenetic perspective. Overall, this manuscript appears to provide meaningful and useful information that extends our knowledge of the range of this important pathogen. I do not have major criticisms, but offer a few comments for consideration:
This manuscript is exceptionally specific to geography and have doubts that the maps represented in Figures 2-7 are worth including. Similarly, Figure 1b is clearly of value, but 1a could be deleted. This should be the author’s choice but something to consider, especially if they have not obtained copyright permissions for those images.
- We had intensive discussions about his topic among the authors and finally decided to keep all maps because we think they are of value for the readers in order to judge the habitat of the virus-positive locations. More so, as we are observing common features over the years, which in the future may help identify natural foci without the previous knowledge of a patient who might have acquired TBEV there. Hence, we would like to keep all the maps in the article. We have obtained copyright permissions for all images.
The subheadings (e.g. 1) Background) do not seem standard for Viruses and should likely be removed.
- We used the template from the Viruses web page and kept the required formatting. We therefore believe that the subheadings are correct. We have noticed that the formatting and naming of sections in the abstract are different from the body of the manuscript, but it is the same in the journal’s template.
Line 225-6: sentence requires editing.
- The sentence has been edited.
- “All isolates from Saxony (from both locations – Sparte am Moor and Waldbad Bernsdorf) have the closest genetic relation to ones from Baden-Württemberg: Wangen and Karsee”
There are some strange mutations in the reference list e.g., line 397 (2. 22. Siuda).
- The formatting of the references has been checked and edited (also ref. nos. 8, 22, 41, 45, 46).